# Electroactive Ultra-Thin rGO-Enriched FeMoO_4_ Nanotubes and MnO_2_ Nanorods as Electrodes for High-Performance All-Solid-State Asymmetric Supercapacitors

**DOI:** 10.3390/nano10020289

**Published:** 2020-02-09

**Authors:** Kugalur Shanmugam Ranjith, Ganji Seeta Rama Raju, Nilesh R. Chodankar, Seyed Majid Ghoreishian, Cheol Hwan Kwak, Yun Suk Huh, Young-Kyu Han

**Affiliations:** 1Department of Energy and Material Engineering, Dongguk University-Seoul, Seoul 04620, Korea; 2Department of Biological Engineering, Inha University, Incheon 22212, Korea

**Keywords:** electrospinning, FeMoO_4_ nanotubes, rGO wrapping, MnO_2_-rGO, asymmetric supercapacitors

## Abstract

A flexible asymmetric supercapacitor (ASC) with high electrochemical performance was constructed using reduced graphene oxide (rGO)-wrapped redox-active metal oxide-based negative and positive electrodes. Thin layered rGO functionality on the positive and the negative electrode surfaces has promoted the feasible surface-active sites and enhances the electrochemical response with a wide operating voltage window. Herein we report the controlled growth of rGO-wrapped tubular FeMoO_4_ nanofibers (NFs) via electrospinning followed by surface functionalization as a negative electrode. The tubular structure offers the ultrathin-layer decoration of rGO inside and outside of the tubular walls with uniform wrapping. The rGO-wrapped tubular FeMoO_4_ NF electrode exhibited a high specific capacitance of 135.2 F g^−1^ in Na_2_SO_4_ neutral electrolyte with an excellent rate capability and cycling stability (96.45% in 5000 cycles) at high current density. Meanwhile, the hydrothermally synthesized binder-free rGO/MnO_2_ nanorods on carbon cloth (rGO-MnO_2_@CC) were selected as cathode materials due to their high capacitance and high conductivity. Moreover, the ASC device was fabricated using rGO-wrapped FeMoO_4_ on carbon cloth (rGO-FeMoO_4_@CC) as the negative electrode and rGO-MnO_2_@CC as the positive electrode (rGO-FeMoO_4_@CC/rGO-MnO_2_@CC). The rationally designed ASC device delivered an excellent energy density of 38.8 W h kg^−1^ with a wide operating voltage window of 0.0–1.8 V. The hybrid ASC showed excellent cycling stability of 93.37% capacitance retention for 5000 cycles. Thus, the developed rGO-wrapped FeMoO_4_ nanotubes and MnO_2_ nanorods are promising hybrid electrode materials for the development of wide-potential ASCs with high energy and power density.

## 1. Introduction

Immense interest in the rapid development of portable flexible electronic tools has promoted the need for lightweight and high-performing energy storage devices to meet the requirement for a resilient power supply [1,2]. Among the electrochemical storage systems, supercapacitors have attracted significant interest as a future power source owing to their high-power densities and exceptionally long cycle lives [3,4]. However, the practical applications of supercapacitors have been generally limited due to their low energy densities. The design of asymmetric supercapacitors (ASCs) with two different sets of electrode systems has provided an efficient strategy for developing the operating voltage window and providing a high energy density to meet the demands of emerging technology [5]. While intensive efforts directed at ASCs have focused on device structures such as V_2_O_5_/carbon [6], MnO_2_/graphene or carbon [7], Co_3_O_4_@C@Ni_3_S_2_/activated carbon (AC) [8], MnO_2_/Fe_3_O_4_ [9], MnO_2_/MoS_2_ [10], CNT-MnO_2_/CNT-SnO_2_ [11], graphene/polypyrrole [12], and graphene–NiOH/graphene [13], most of the assembled supercapacitor devices have been configured with liquid electrolytes. Moreover, most of the ASC devices have been configured with pseudocapacitive materials as the positive electrode while carbon-based materials have commonly been used as negative electrodes [5,14]. Until now, intensive research has been focused on exploring the positive electrodes, while negative electrode materials were rarely studied apart from the carbon-based electrodes. Generally, ASCs using carbon-based negative electrodes have suffered due to the low energy density with a lower capacitance of the activated carbon (AC) in the aqueous electrolytes, especially in neutral conditions. Moreover, in comparison with the electrode surface features, the flexible electrode has held many advantages along with high flexibility and stability [15]. At this point, the flexible ASCs are expected to attract a huge demand, with their flexibility providing the key to portable energy storage systems.

Considerable attention has focused on the advantages and durability of Fe- and Mo-based electrodes for ASCs. In particular, the MoO-based electrode has attracted interest as a negative electrode system due to its low electrical resistivity, multiple valance states, rich chemistry, natural abundance, and affordable cost [16]. The incorporation of MoO_3_ with carbon results in a specific capacitance of 413.08 F g^−1^ at 8 A g^−1^ due to the great intercalating ability of MoO_3_ which promotes the large current density [17]. The MoO_2_/CC-based negative electrode in NaMnO_2_/CC//MoO_2_/CC ASCs exhibits an energy density of 0.92 mW h cm^−3^ and a maximum volume capacitance of 2.04 F cm^−3^ along with outstanding cycling stability with 97.33% capacitance retention even after 6000 cycles [18]. The construction of graphene-induced MoO_3_ nanosheets as a negative electrode supported the short diffusion path and promoted the electron transport pathway. In addition, the inclusion of graphene on both electrode surfaces provided the hybridized interlayers with an extra interface and promoted the pseudocapacitive reaction and rate capability [19]. With the graphene-promoted electrode surface, the GrMnO_2_/GrMoO_3_ ASC device exhibited a specific capacitance of 307 F g^−1^ with an energy density of 42.6 W h kg^−1^ at a power density of 276 W kg^−1^. In addition, the Fe_2_O_3_-based electrode systems have received significant attention because of their multiple valance states, rich redox chemistry, high specific capacitance, functional capability, and high stability as negative electrodes, high theoretical capacitance, environmentally friendly properties, and natural abundance, which make them suitable for commercialization. The design of Fe_2_O_3_ nanotubes as the negative electrode in MnO_2_/Fe_2_O_3_ ASCs results in excellent stability with a wide potential window of 0.0–1.6 V along with an outstanding energy density [20]. The richness of the Fe phase has been shown to promote the redox-active hydroxides in the electrode surface [21]. The construction of nanostructured electrode materials with tubular assemblies has significant advantages, including high surface area, effective heat transfer ability and a favorable electrode–electrolyte interface [22,23,24,25,26]. Decorating carbon nanotube (CNT) sponges with Fe_2_O_3_ horns to generate a porous electrode has resulted in a high specific capacitance of 296 F g^−1^ owing to the combined influence of the electric double-layer capacitance (EDLC) of the CNTs and the pseudocapacitance of Fe_2_O_3_ [27]. The inclusion of the ternary FeMoO_4_ network has demonstrated the advantage of promoting the individual functional properties to provide a high-performing electrode. The design of molybdenum-containing transition metal oxides has been promoted as an advanced anode material through the high oxidation of molybdate [27]. The high theoretical capacitance of 992.3 mA h g^−1^, attributed to the high valance of Fe^3+^ and Mo^6+^, has prompted interest in FeMoO_4_-based electrodes as the anode material in ASCs. The reduced graphene oxide (rGO)–FeMoO_4_ nanocomposite has provided a specific capacitance of 135 F g^−1^ and the promotional impact of rGO was found to induce lower electrochemical resistance and a longer cycle life [28]. Electrospinning is a feasible approach for designing one-dimensional fiber assembly even in binary or ternary compositions [29,30]. Performing experiments using polymerics with different physical parameters led to control of the morphology of the electrospun fibers using the effects of external or post-thermal treatments [31,32]. Tuning the surface area by controlling the morphology and inducing a highly conductive functional network has enhanced the potential of the heterostructural electrode system.

In the present work, the heterostructural rGO/FeMoO_4_ hybrid nanotube network was constructed via electrospinning followed by a hydrothermal technique. The fabrication of tubular FeMoO_4_ NFs with the twin-walled rGO functionalities results in the design of a hybrid network with a high surface area. Benefiting from the tubular assembly and the layered nanostructures of FeMoO_4_ and rGO flakes, the prepared FeMoO_4_/rGO electrodes exhibited superior electrochemical performances with a high specific capacitance of 135.2 F g^−1^ at 2 A g^−1^. Considering the advantages of the FeMoO_4_/rGO electrode material, a flexible ASC was fabricated using polyvinyl alcohol (PVA)/Na_2_SO_4_ electrolyte with FeMoO_4_/rGO and MnO_2_/rGO as negative and positive electrodes, respectively. For the first time, new ASC systems based on MnO_2_ and FeMoO_4_ were fabricated using a cost-effective, facile and scalable approach. In particular, the design of the tubular FeMoO_4_ NFs with highly conductive ultra-thin rGO features resulted in efficient electrochemical responses of the active surfaces due to the high internal surface area and short diffusion path of the hybrid electrode surface. The MnO_2_ nanorods were directly grown on the carbon cloth (CC) which facilitates the effective electrical mobility. The CC serves as a flexible current collector and as a scaffold for the heterostructural network. The ASC assembly of rGO–FeMoO_4_/rGO–MnO_2_ displayed a specific capacitance of 54.4 F g^−1^ at 2 A g^−1^ and delivered a wide voltage window of 0.0–1.8 V. With the advantage of a high-performing negative electrode with a wide potential window, the assembled ASC device exhibited a high energy density of 38.8 W h kg^−1^ at a power density of 1344.5 W kg^−1^ and a high capacitance retention of 93.37%. This work demonstrates that the FeMoO_4_/rGO hybrid electrodes have extended the voltage window compared to the carbon-based negative electrodes and have a broad potential application in flexible ASCs.

## 2. Materials and Methods

### 2.1. Preparation of rGO Ultra-Thin Nanoflakes

Two-dimensional (2D) well-dispersed graphene oxide (GO) was synthesized via the Hummers method [33]. A 3:1 mixture of H_2_O_2_ and NH_3_OH (40 mL) was added dropwise to a homogeneous dispersion of GO (60 mL, 1 mg mL^−1^) in a round-bottomed flask and reflexed at 200 °C for 24 h. Finally, a well-dispersed ultra-thin rGO solution was obtained. 

### 2.2. Preparation of FeMoO_4_/rGO Hybrids

The FeMoO_4_ nanotubes were prepared via a facile electrospinning process followed by thermal annealing. In a typical process, 1.3 g of polyacrylonitrile (PAN), poly(methyl methacrylate) PMMA (8:2) in DMF (10 mL) was stirred for 3 h to form the homogeneous precursor solution. Further, FeCl_3_ (20 wt.% with polymeric source) and MoCl_3_ (10 wt.% with polymeric source) were added to the polymeric solution and stirred overnight to attain a homogeneous clear solution. The polymeric sol was then loaded into a plastic syringe with a 23-gauge injection needle, which was connected to the high-voltage power supply (Nano EC). Under the flow rate of 1 mL h^−1^ with 12 kV of applied voltage, the electrospun fibers were collected in the aluminum foil-covered drum collector which was maintained at a distance of 15 cm from the needle. The as-spun Fe- and Mo-loaded PAN/PMMA composite NFs were calcinated at 500 °C for 3 h in the air to yield the pale reddish hollow FeMoO_4_ NFs. The effective decomposition of PMMA created the tubular morphology, while the later decomposition of PAN provided a 1D scaffold with a fibrous morphology. For comparison, the electrospinning was also performed without PMMA in the precursor solution to obtain the solid NFs. A 0.3 g mass of as-prepared NFs was subjected to the surface functionalization using (3-aminopropyl)trimethoxysilane (ATS, 1 mL) in isopropyl alcohol (IPA, 60 mL) at 60 °C overnight, and the unattached silane functionalities were removed by multiple centrifugations. The ultra-thin rGO nanoflakes were coated onto the FeMoO_4_ NF surface via the facile hydrothermal technique. First, a well-dispersed 30 mL of rGO solution (0.6 mg/mL) was dissolved in 30 mL of deionized (DI) water and 0.3 g of hollow FeMoO_4_ NFs were then dispersed in the solution under mild stirring. Finally, the solution was transferred to the Teflon-lined autoclave and heated at 140 °C for 6 h. Under the thermal treatment, the well-dispersed ultrathin rGO was functionally attached to the hollow FeMoO_4_ nanofibrous surface and the strong bonding interaction with the functionalized FeMoO_4_ NFs was favored. The precipitated rGO–FeMoO_4_ NFs were centrifuged, washed three times with DI water and ethanol, and vacuum dried at 50 °C for 12 h.

### 2.3. Preparation of MnO/rGO Hybrids

The cleaned carbon cloth (CC) with dimensions of 2 × 1 cm (4 numbers) were immersed in aqueous KMnO_4_ solution (35 mL, 0.84 mg/mL) and dispersed in 15 mL of rGO solution (0.6 mg/mL) was then added dropwise under constant stirring. The homogeneous aqueous dispersion was obtained via the hydrothermal process in a Teflon-lined stainless-steel container which was sealed and maintained at 160 °C for 12 h. After cooling down, the CC was collected and cleaned with DI water and dried at 110 °C for 12 h in a vacuum. For comparison, un-modified MnO_2_ (without rGO) was also prepared on the CC.

### 2.4. Characterization Methods

The structural analyses of the as-prepared MnO_2_ and FeMoO_4_ fibers were performed by powder X-ray diffraction (X’Pert-PRO MRD, Philips, The Netherlands). The morphologies of the active materials were examined by high-resolution field emission microscopy (HRSEM, Hitachi, Japan) and high-resolution transmission electron microscopy (FETEM, JEM-2100F, JEOL, Japan), and the elemental compositions were analyzed by energy-dispersive X-ray spectroscopy (EDX, Oxford Instruments, High Wycombe, UK). The structural compositions and electronic states were analyzed by X-ray photoelectron spectrometry (XPS, Thermo Fisher Scientific, K-Alpha, USA) with non-monochromatic Al K-*α* radiation (1486.6 eV). The Brunauer–Emmett–Teller (BET) specific surface area (*S*_BET_) was measured by the multipoint BET method (ASAP ZOZO, Micromeritics, GA, USA).

### 2.5. Electrochemical Measurement

The FeMoO_4_ NF-based working electrode was prepared by mixing the electroactive NFs, carbon black and polyvinylidene fluoride (PVDF) binder in a mass ratio of 80:10:10 with N-methyl pyrrolidinone (NMP) and grinding to form a homogeneous slurry. The resulting slurry was coated onto the CC (1 × 1 cm^2^), followed by drying at 70 °C for 12 h in a vacuum oven. As the MnO_2_ was directly grown on the carbon fabric, it was utilized as a binder-free electrode. All the electrochemical measurements were performed at room temperature on the Metrohm Auto lab workstation (PGSTAT302N). The Ag/AgCl electrode and platinum foil were used as a reference and counter electrodes, respectively, in a three-electrode cell and 1 M Na_2_SO_4_ used as an aqueous electrolyte. Electrochemical tests such as cyclic voltammetry (CV) and galvanostatic charge–discharge (GCD) were performed at various scan rates and current densities between −0.8 and 0 V and between 0 and 1 V vs. Ag/AgCl for the rGO/FeMnO_4_ and rGO/MnO_2_ electrodes, respectively. Electrochemical impedance spectroscopy (EIS) was performed with the potential amplitude of 5 mV over the frequency range of 100 kHz to 0.01 Hz. The specific capacitance was calculated from the GCD curve according to the equation:$$C=\frac{I\;\Delta t}{m\;\Delta V}$$
where *C* is the specific capacitance (F g^−1^), Δ*t* is the total discharge time (s), I is the constant discharge current (A), m is the mass of the active material in the electrode (g), and Δ*V* is the applied potential window (V).

### 2.6. Fabrication of the All-Solid-State ASC Device 

The all-solid-state flexible ASC device was fabricated using rGO/MnO_2_@CC as the positive electrode and rGO/FeMoO_4_@CC as the negative electrode in the PVA/Na_2_SO_4_ gel electrolyte. The PVA/Na_2_SO_4_ gel was prepared by mixing 2 g of PVA and 0.5 M Na_2_SO_4_ in deionized (DI) water (20 mL) and heating at 85 °C with constant stirring until the mixture became clear. The gel electrolyte was first coated uniformly onto the rGO/MnO_2_@CC and rGO/FeMoO_4_@CC electrodes and dried for 2 h in the open air. Finally, the ASC device was fabricated by combining the two electrodes with the gel electrolyte and the device assembly was packed with insulting tape and dried overnight in a hood at room temperature. The loading mass of the active electrode materials was set according to the following equation:$$m^{+}/m^{-}\;=\;(C^{-}\;\times \;\Delta V^{-})/(C^{+}\;\times \;\Delta V^{+})$$
where *C* is the specific capacitance, Δ*V* is the potential window, the + and − indicate the positive and negative electrode, respectively, and m is the mass of the active electrode material. The energy density (*E*) for the ASC was calculated according to E = CV^2^/(2 × 3.6), while the power density (*P*) was evaluated according to P = E/Δ*t* (where Δ*t* is the discharge time in GCD). 

## 3. Results

### 3.1. Constructing the Negative Electrode Material

Scheme 1 presents the growth strategy for preparing the tubular FeMoO_4_ NFs with the ultra-thin rGO functionalities. The co-polymeric precursors PAN and PMMA used for electrospinning with the Fe and Mo sources induce the tubular morphology by means of their two distinct decomposition temperatures. The as-spun FeMo–PAN/PMMA fibers were thermally decomposed at 500 °C for 3 h to prepare the tubular FeMoO_4_ NFs. In the second step, the thermally reduced ultra-thin GO was functionalized on the tubular FeMoO_4_ NFs via the hydrothermal process to improve the surface functionality of the NFs. Appendix A presents the SEM images of the as-spun FeMo-PAN and FeMo-PAN/PMMA, indicating that the obtained NFs had diameters of 450 nm and 480 nm, respectively. Figure 1 shows the structural morphologies of the tubular FeMoO_4_ and rGO/FeMoO_4_ NFs indicated by the SEM and TEM analyses. The SEM image in Figure 1a shows that the FeMoO_4_ NF synthesized using the PAN scaffold has a fiber diameter of 250 nm with a solid core. The high-resolution SEM image in Figure 1b shows the dense grain structure of the FeMoO_4_ fiber surface with quite smooth surface features, whereas the addition of PMMA/PAN as a scaffold in the spinning process (Figure 1c) results in the formation of tubular FeMoO_4_ NFs having diameters of around 250 nm. The high-resolution image in Figure 1d shows the tubular FeMoO_4_ NF with a shell wall thickness of 20 nm. The inclusion of PMMA with the PAN scaffold leads to earlier thermal decomposition which favors the formation of the tubular structure, while the PAN network promotes the stability of the fibers during the annealing process. When PMMA is used alone as the scaffold, aggregation of the FeMoO_4_ nanograins leads to the complete collapse of the fibrous morphology after annealing (Appendix A). Figure 1e,f present the SEM images of the rGO-functionalized FeMoO_4_ tubular NFs which indicate the hybrid structure of an ultrathin rGO layer covering the tubular NFs and the wrinkled rGO on the FeMoO_4_ NF surface. Further, rGO functionality on the FeMoO_4_ tubular NFs was confirmed through the Raman scattering spectra. Appendix A results in the peaks at 816, 904 and 949 cm^−1^ are assigned to FeMoO_4_ [28,34]. In addition, with the completion of rGO wrapping, rGO–FeMoO_4_ NFs displayed additional peaks at 1360 and 1609 cm^−1^, representing the D and G bands of graphene, thereby confirming the wrapped features of rGO along with the FeMoO_4_ NF [35]. The ultrathin rGO layer is tightly bonded with the FeMoO_4_ surface and promotes the stability and surface conductivity during electrolyte ion diffusion and enhances the electrochemical performance. The tubular structure promotes the rGO loading on the inner and outer surfaces of the tubular features, leading to the twin shell wall effect on the hybrid fibrous surface. The pristine thermally reduced rGO (Appendix A) consists of a few layers of rGO flakes with sizes in the range of 40 to 80 nm, which facilitate binding to the inner and outer walls of the FeMoO_4_ tubular NFs. Through the ATS-based surface functionalization, the rGO layers were selectively bound to the fiber surface without any aggregation.

The structural and morphological properties of the rGO–FeMoO_4_ tubular NFs were further examined by TEM and HRTEM analyses. The TEM images of the rGO nanoflakes (Appendix A) indicate the ultra-thin layered features, while the TEM images of the bare FeMoO_4_ nanotubes indicate the nanograin features of the tubular wall (Appendix A). The high magnification TEM images of the rGO–FeMoO_4_ nanotubes are presented in Figure 1g,h, the latter image showing the heterostructure of well-dispersed surface-functionalized ultrathin rGO layers on the tubular wall of the FeMoO_4_ NFs. The high-resolution TEM images of the tubular rGO–FeMoO_4_ NFs in Figure 1h also shows the decoration of ultrathin rGO layers on the tubular wall with a size of 15 nm. The HRTEM results showed that the nanograins had a lattice spacing of 0.316 nm in the (220) crystal plane of the FeMoO_4_ and that the rGO functionalization on the tubular wall resulted from layer formation over the nanograins (Appendix A). The structural crystallinity of the tubular fiber was investigated through the selected area electron diffraction (SAED) pattern (inset of Figure 1g), in which the diffuse ring pattern corresponds to the carbon features and the dots represent the crystalline nature of the FeMoO_4_ NF. The EDX results demonstrate the presence of Fe, Mo, O, and C on the fibrous surface (Appendix A). The corresponding elemental mapping (Figure 1h) confirms that the elemental distribution of C, Fe, Mo, and O on the fibrous surface is homogeneous and that the presence of the carbon features results in the rGO surface functionality and the decomposed PAN network in the form of carbon over the fibrous surface. 

Figure 2a presents the XRD pattern of the rGO-functionalized FeMoO_4_ nanofibers, which is well-indexed to the monoclinic phase of β-FeMoO_4_ (Joint Committee on Powder Diffraction Standards (JCPDS) card No. 01-89-2367). The additional broad peak at 26° on the rGO/FeMoO_4_ nanofibers next to the FeMoO_4_ indicates the surface functionality of rGO on the FeMoO_4_ surface. The trace appearance of cubic Fe_3_O_4_ (JCPDS card No: 65-3107) indicates the richness of Fe ions on the tubular fiber surface which results in faster oxidation to form Fe_3_O_4_ on the fiber surface. The survey XPS spectrum confirmed the existence of Fe, Mo, C, and O on the fiber surface (Figure 2b). The Fe 2p core-level XPS spectrum (Figure 2c) indicates the presence of mixed oxidization states of Fe^2+^ (711.1 eV and 724.9 eV) and Fe^3+^ (713.3 eV and 727.6 eV) with the satellite peak at 718.2 eV [36]. Figure 2d indicates the high-resolution Mo 3d with two prominent peaks at 232.5 eV and 235.7 eV due to the binding energies of Mo 3d_3/2_ and Mo 3d_5/2_, respectively, corresponding to the Mo^6+^ oxidization states in FeMoO_4_ [37,38]. The high-resolution C 1s spectra were deconvoluted (Figure 2e) with a C–C vibration at 284.6 eV, oxygen-containing vibrations at 286.5 eV (C–O) and 288.7 eV (O−C=O), and the trace of C–N interactions at 285.7 eV [39]. The high-resolution O 1s spectra were deconvoluted into two peaks at 530.3 eV and 531.8 eV representing the lattice oxide vibration and chemisorbed oxygen with Fe and Mo ions, respectively [40]. The nitrogen adsorption–desorption isotherm was investigated to explore the specific surface area and the pore size distribution of the rGO-loaded FeMoO_4_ tubular nanofibers (Appendix A). The tubular fibers resulted in the BET specific surface area of 5.64 m^2^ g^−1^ and the isotherm demonstrates a type IV hysteresis loop, suggesting a mesoporous structure. The pore size distribution was calculated to be about 27.1 nm according to the Barrett–Joyner–Halenda (BJH) method, which confirms the mesoporous structure. 

Figure 3a presents the CV curve of the FeMoO_4_ and rGO–FeMoO_4_ hybrid electrodes within the potential range of −0.8 to 0 V (vs. Ag/AgCl) at the scan rate of 10 mV s^−1^. The hybrid electrodes displayed noticeable redox peaks and demonstrated that, with the rGO feature, the FeMoO_4_ electrodes possessed larger integral areas than the pristine FeMoO_4_ tubular fibers and, hence, the higher specific capacitance (Appendix A). Figure 3b presents the CV curves of the tubular rGO–FeMoO_4_ electrodes at various scan rates ranging from 1 to 200 mV s^−1^. Even at the high scan rate, the CV curve of the rGO–FeMoO_4_ electrode maintained its typical shape, indicating the excellent rate capability. The GCD curve results indicate that, at the various scan rates within the potential window of −0.8 V to 0 V, the rGO–FeMoO_4_ electrode displays high coulombic efficiency and ideal capacitance behavior with a superior charge storage rate. Compared with the pristine tubular FeMoO_4_ electrode, the rGO–FeMoO_4_ electrodes displayed specific capacitances at similar current densities, comparable to the above CV results. However, there is only a small IR drop of 0.028 V at a high current density of 20 A g^−1^ (Figure 3d), indicating the excellent electrical conductivity and efficient mobility of electrolyte ions. Figure 3e indicates the calculated specific capacitance of the hybrid rGO–FeMoO_4_ electrode at various current densities from the GCD curve. The rGO–FeMoO_4_ hybrid electrode displayed a specific capacitance of 135.2 F g^−1^ at 2 A g^−1^, which is much greater than that of FeMoO_4_ (32.6 F g^−1^) at the same current density, and remains as high as 100.3 F g^−1^ even at the high current density of 20 A g^−1^, thus demonstrating good rate capability. The synergistic effect between the thin-layered rGO and FeMoO_4_ NFs increases the specific capacitance of the rGO-wrapped FeMoO_4_ electrode. Further, the rGO acts as a conductive channel and an active interface on the FeMoO_4_ nanofibers. The rGO–FeMoO_4_ hybrid electrodes with various rGO loadings have been compared (Appendix A) for the investigation. However, while the efficient loading of the rGO (60% and 20%) resulted in good rate performances as compared to others, their specific capacitances (82.5 F g^−1^ and 127.5 F g^−1^) were much lower than for the 40% loading of rGO on the FeMoO_4_ fibers (135.2 F g^−1^). Further, the 80% loading of rGO resulted in the lower specific capacitance of 57.5 F g^−1^. Highly-loaded rGO on FeMoO_4_ may completely arrest the active sites through high surface capping, which would restrict the utility of FeMoO_4_ in the hybrids. The electrochemical impedance spectroscopy (EIS) spectra in Figure 3f indicates the fundamental behavior of the electrode surface material. The high-frequency semicircle indicates the internal charge transfer resistance and the vertical line in the low-frequency region indicates the diffusion limit of the rGO-FeMoO_4_ hybrid electrodes. The Nyquist plot in Figure 3f shows that the rGO–FeMoO_4_ hybrid has a much higher phase angle in the lower frequency range, indicating the ideal capacitive nature and low charge transfer resistance of 1.73 Ω due to the fast ion diffusion via the tubular surface with the high surface conductive skeleton which accelerates the electrolyte ionic transport. In Figure 3g, the outstanding electrochemical stability of the rGO–FeMoO_4_ electrodes is indicated by the 3.55% loss of initial specific capacitance and 98.34% of its coulombic efficiency after 5000 cycles. The inset in Figure 3g, the overlapping function of CV and GCD curves, indicates excellent stability before and after the cycling stability and EIS. The cyclic stability results in a slight increase in charge transfer resistance of 0.11 Ω (*R_s_*). The exceptional stability and promising electrochemical performance of the rGO–FeMoO_4_ electrodes were ascribed to the following factors: (i) the bimetallic features of Fe and Mo induced more redox sites to promote the electrochemical response of the fiber network; (ii) the integral configuration of rGO bound to the FeMoO_4_ surface decreased the internal charge resistance and promoted the efficient ion diffusion channel with the proper utilization of active material; and (iii) the tubular morphology provided the highly exposed surface active sites with shortened ionic diffusion.

### 3.2. Constructing the Positive Electrode Material

To fabricate the cathode material, rGO–MnO_2_ nanorods were grown directly on the CC via a facial hydrothermal process at 160 °C. The results of the SEM and TEM morphological analyses of the rGO–MnO_2_ are presented in Figure 4. As a binder-free electrode, the MnO_2_ nanoflakes were effectively grown on the CC as a hierarchical platform with self-stacked structures (Figure 4a,b). The composite features were more clearly characterized by the TEM analysis (Figure 4c,d). It can be observed that the individual nanoflakes were tightly tagged with the rGO layers on their surface. The lattice spacing of the MnO_2_ nanoflakes was around 0.69 nm, representing the (001) crystal planes of Birnessite MnO_2_ (Figure 4e) [41]. The SEAD pattern (Figure 4f) reveals the polycrystalline nature. In the XRD spectrum (Figure 4g) the broad band at 26.4° corresponds to the carbon substrate (CC) and the other diffraction peaks correspond to tetragonal α-MnO_2_. The trace appearance of a broad peak at 26.4° in the rGO–MnO_2_ samples results from the rGO oxide features on the electrode surface. The XPS survey spectrum of the rGO–MnO_2_ presented in Figure 4h indicates the existence of the elements Mn, O and C. The high-resolution Mn 2p spectrum (inset of Figure 4h) displays two prominent peaks at 641.6 eV and 653.4 eV separated by 11.8 eV, confirming the presence of MnO_2_ in the composite [42].

Figure 5a presents the CV curves of the hydrothermally grown free-standing MnO_2_ and rGO/MnO_2_ electrodes at a scan rate of 50 mV s^−1^. The rGO features on MnO_2_ promote the redox signal under the applied potential window. The CV curve of the rGO–MnO_2_ electrode at various scan rates from 1 to 200 mV s^−1^ (Figure 5b) within the potential window of 0–1 V reveals the typical rectangular profile with an excellent rate capability. Figure 5c presents the GCD curves of the rGO–MnO_2_ free-standing electrode at various current densities from 1 to 20 A g^−1^ within the potential window of 0–1 V. The GCD curves of the rGO–MnO_2_ electrode are highly linear with high coulombic efficiency, indicating ideal capacitive character. Thus, the rGO–MnO_2_ electrode displays high coulombic efficiency in a wide potential window along with the superior specific capacitance of 324.5 F g^−1^ at 1 A g^−1^ (Figure 5d), which is much more efficient than the pristine MnO_2_ (232.3 F g^−1^). The Nyquist plot (Figure 5e) displays a semicircle (*R_s_* = 3.28 Ω) in the high-frequency region and a vertical plot in the low-frequency region, indicating the excellent rate capability. Figure 5f indicates the cycling stability of the rGO–MnO_2_ free-standing electrode with the capacitance retention of 90.26% after 5000 cycles at the current density of 10 A g^−1^. 

### 3.3. Construction of the rGO–FeMoO_4_@CC/rGO–MnO_2_@CC ASC Device

In view of the excellent electrochemical performances of the positive (rGO–MnO_2_) and negative (rGO–FeMoO_4_) electrodes in the aqueous electrolyte, the flexible ASC device was fabricated using rGO–MnO_2_ as the positive electrode and rGO–FeMoO_4_ as the negative electrode and Na_2_SO_4_/PVA solid gel electrolyte. The assembled device is depicted in Figure 6a. The mass loading of the ASC device was balanced by following the relationship of q^+^ = q^−^ prior to assembly. Figure 6b shows the working potential windows of both the rGO–MnO_2_ and rGO–FeMoO_4_ electrodes at the scan rate of 30 mV s^−1^ with the Na_2_SO_4_-based electrolyte. With the advantages of a wide potential window and high specific capacitance, the ASC device was expected to operate at potentials of up to 1.8 V. Under the fixed scan rate, Figure 6c presents the CV curves of the rGO–FeMoO_4_@CC/rGO–MnO_2_@CC ASC within various potential windows ranging from 0–0.8 V to 0–2.2 V (Figure 6c). The rGO–MnO_2_@CC/rGO–FeMoO_4_@CC ASC device provides evidence of synergistic electrochemical performances up to the potential window of 0–1.8 V. The typical CV curve (Figure 6d) of the rGO–MnO_2_@CC/rGO–FeMoO_4_@CC ASC device at various current densities within the potential window of 0–1.8 V demonstrates the promising pseudocapacitive storage behavior even at high current density. 

Figure 6e presents the galvanic charge device (GCD) curve of the rGO–MnO_2_@CC/rGO–FeMoO_4_@CC ASC device at various current densities within the potential window of 0–1.8 V, which clearly indicates highly symmetric curves with a high coulombic efficiency (>95%) for all the respective current densities, thereby demonstrating the excellent reversible efficiency. The specific capacitance values of the rGO–MnO_2_@CC/rGO–FeMoO_4_@CC ASC estimated from the GCD curve are 57.7, 54.4, 53.4, 52.1, 50.2, 44.4, and 44.1 F g^−1^ at the respective current densities of 1, 2, 3, 4, 5, 10, and 20 A g^−1^. The loss in specific capacitance at high current density is ascribed to the increase in the internal resistance of the hybrid device assembly. Interestingly, the rGO–MnO_2_@CC/rGO–FeMoO_4_@CC ASC devices retained 76.4% of their initial capacitance on increasing the current density by 20 times, demonstrating the high rate capability of the ASC electrode surface. In order to understand the electron/ion transport performance, Figure 6f presents the Nyquist EIS spectra of the rGO–MnO_2_@CC/rGO–FeMoO_4_@CC ASC electrode surface. The internal resistance (*R_s_*) was calculated as 6.72 Ω and the internal charge transfer resistance was calculated as 1.13 Ω from the semicircle in the high-frequency region. The results of the cyclic stability tests for the rGO–MnO_2_@CC/rGO–FeMoO_4_@CC ASC devices indicated that about 93.37% of the primary capacitance was retained after 5000 GCD cycles at the current density of 4 A g^−1^ (Figure 6h). The SEM images presented in Appendix A indicates that the rGO–FeMoO_4_ NFs retained their morphology after the cyclic test, thus demonstrating their potential stability. The obtained results indicated the superior electrochemical performances of the rGO–MnO_2_@CC/rGO–FeMoO_4_@CC ASC devices along with long-term stability and a wider potential window than the previous reports summarized in Appendix A.

Figure 7a presents a Ragone plot of the fabricated ASC device derived using the GCD results. The rGO–MnO_2_@CC/rGO–FeMoO_4_@CC device exhibited outstanding energy of 38.8 W h kg^−1^ at the power density of 1344.5 W kg^−1^. Further, at the high-power density of 26,872.9 W kg^−1^, the ASC hybrid device maintained an energy density of 29.9 W h kg^−1^, thus demonstrating an outstanding rate performance. In a comparative investigation, the energy and power densities of the rGO–MnO_2_@CC/rGO–FeMoO_4_@CC ASC were compared with those of recently reported ASCs such as Co_3_O_4_@MnO_2_/MEGO (17.7 W h kg^−1^ at 158 W kg^−1^) [43], ZnCo_2_O_4_@MnO_2_/AC (29.41 W h kg^−1^ at 628.4 W kg^−1^) [44], ZnCo_2_O_4_@Ni*_x_*Co_2*x*_(OH)6*_x_*/AC (26.2 W h kg^−1^ at 511.8 W kg^−1^) [45], CaMoO_4_/AC (18.7 W h kg^−1^ at 362 W kg^−1^) [46], MnO_2_/Fe_2_O_3_ (53.55 W h kg^−1^ at 1280 W kg^−1^) [47], MnO_2_–GNS/FeOOH–GNS CNTs (30.4 W h kg^−1^ at 237.6 W kg^−1^) [48], MnO_2_/FeOOH (24 W h kg^−1^ at 450 W kg^−1^) [49], and MnO_2_ nanowire/graphene (30.4 W h kg^−1^ at 100 W kg^−1^) [50] (Figure 7a and Appendix A). A photographic image of three rGO–MnO_2_@CC/rGO–FeMoO_4_@CC ASC devices connected in series to light two red LEDs are presented in Figure 7b. The operation of the flexible ASC device under different bending positions, and the respective CV curves, are shown in Figure 7c. The CV curves of the rGO–MnO_2_@CC/rGO–FeMoO_4_@CC ASC with various bending angles are almost identical to that in the absence of any deformation.

The photographs of the ASC device in various bending positions demonstrate the high flexibility of the rGO–MnO_2_@CC/rGO–FeMoO_4_@CC ASC device. The obtained results demonstrated the fabrication of a high-performing flexible ASC device as an integrated hybrid smart textile. The notably promising performances of the rGO–MnO_2_@CC/rGO–FeMoO_4_@CC ASC device can be ascribed to the following causes: (i) the promotion of electroactive sites by the tubular and layered features of rGO-bonded FeMoO_4_ nanostructures, and (ii) the extended potential window of the electrode surface provided by the unique construction of rGO–FeMoO_4_ and rGO–MnO_2_ on the flexible CC. It is, therefore, demonstrated that the rGO–MnO_2_@CC/rGO–FeMoO_4_@CC ASC device fabricated at low cost holds great promise for highly efficient energy storage in flexible modern devices.

## 4. Conclusions

In summary, we have demonstrated the fabrication of a flexible hybrid rGO–FeMoO_4_@CC/rGO–MnO_2_@CC-based all-solid-state ASC device via a low-cost, simple strategy using electrospinning and hydrothermal processes. The as-prepared rGO-bonded tubular FeMoO_4_ NFs possessed a specific capacitance of 135.2 F g^−1^ due to their mixed oxide states with a high surface area. Tagging the ultrathin rGO layers onto the active material surface provided a surface-conductive channel, further increasing the effective path for electron transport and promoting the morphological stability of the active materials. The fabricated rGO–MnO_2_@CC/rGO–FeMoO_4_@CC ASC using rGO–MnO_2_@CC as the positive electrode and rGO–FeMoO_4_@CC as the negative electrode exhibited a specific energy density of 38.8 W h kg^−1^ with efficient cycling stability and excellent rate capability. These results deliver valuable insights on the fabrication of electrode materials with high surface-active sites using the rGO-based hybrid platform to increase the potential window and specific energy density of ASCs.

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
