# Peer review of "Electroactive Ultra-Thin rGO-Enriched FeMoO4 Nanotubes and MnO2 Nanorods as Electrodes for High-Performance All-Solid-State Asymmetric Supercapacitors"

_nanomaterials, 2020, doi:10.3390/nano10020289_

Round 1

Reviewer 1 Report

The manuscript has novel elements and well presented. There are some minor concerns that I invite authors to modify the followings:
1- The english needs to be improved a bit. There are some big sentences that is difficult to follow.
2- there are only 2-3 equations in the text and it has some question marks ??in it!!
3- experimental details, validations, error analysis needs to be better presented.
4- Some figures such as 6c, 6d, 6e, etc. are a bit difficult to read. In black and white print impossible to distinguish. 

5- also add some paper regarding the properties and applications of nanotubes in general. some examples are :

https://www.sciencedirect.com/science/article/abs/pii/S0017931017314783

https://www.sciencedirect.com/science/article/abs/pii/S0735193315001682

https://www.sciencedirect.com/science/article/pii/S0378437119300536

https://link.springer.com/article/10.1007/s10973-019-08838-w

Reviewer 2 Report

General: Please check the grammatical errors, wherever required. English needs to be refined/polished throughout the manuscript. General: Authors are requested to unify the font usage. General: I suggest authors to use one / instead of //, while mentioning device structure for positive electrode. Similar occurrences throughout the manuscript needs to be fixed from Introduction to Conclusions. Introduction: The striking advantages of using nanotubes and nanorods as electrodes need to be clarified. Certain chemicals (PVA, PAN, PMMA etc.) need to be expanded before using abbreviations. I suggest authors to cite articles pertaining to electrospinning and conjugated polymers (Progress in Polymer Science 88, 1-129, 2019, Rev. 2019, 119, 8, 5298-5415, 2019, Materials Chemistry and Physics 154, 125-136, 2015, Catalysts 9 (2), 170, 2019) R n D: All equations need to be revisited. Really impressed with the data collection for material characterization and electrochemical performance evaluation. Well done. Figure 2. Mention core-level spectrum for (c)-(f) What about CV in Figure 3(b) after 200 mV/s ? Similarly in 5(b)
